# Regional-Scale Monitoring of Wheat Stripe Rust Using Remote Sensing and Geographical Detectors

Mingxian Zhao [1,2,3], Yingying Dong [1,2,3,*], Wenjiang Huang [1,2,3] , Chao Ruan [4] and Jing Guo [1,2,3]

1   Key Laboratory of Digital Earth Science, Aerospace Information Research Institute, Chinese Academy of Sciences, Beijing 100094, China; zhaomingxian21@mails.ucas.ac.cn (M.Z.); huangwj@aircas.ac.cn (W.H.); guojing211@mails.ucas.ac.cn (J.G.)
2   International Research Center of Big Data for Sustainable Development Goals, Beijing 100094, China
3   University of Chinese Academy of Sciences, Beijing 100049, China
4   School of Internet, Anhui University, Hefei 230601, China; ruanchao@aircas.ac.cn
*   Correspondence: dongyy@aircas.ac.cn; Tel.: +86-10-82178178

**Abstract:** Realizing the high-precision monitoring of wheat stripe rust over a large area is of great significance in ensuring the safety of wheat production. Existing studies have mostly focused on the fusion of multi-source data and the construction of key monitoring features to improve the accuracy of disease monitoring, with less consideration for the regional distribution characteristics of the disease. In this study, based on the occurrence and spatial distribution patterns of wheat stripe rust in the experimental area, we constructed a multi-source monitoring feature set, then utilized geographical detectors for feature selection that integrates the spatial-distribution differences of the disease. The research results show that the optimal monitoring feature set selected by the geographical detectors has a higher monitoring accuracy. Based on the Random Forest (RF), eXtreme Gradient Boosting (XGBoost), and Support Vector (SVM) models, the disease monitoring results demonstrate that the monitoring feature set constructed in this study has an overall accuracy in its disease monitoring that is 3.2%, 2.7%, and 4.3% higher, respectively, than that of the ReliefF method, with Kappa coefficient higher by 0.064, 0.044, and 0.087, respectively. Furthermore, the optimal monitoring feature set obtained by the geographical detectors method exhibits a higher stability, and the spatial distribution of wheat stripe rust in the monitoring results generated by the different models demonstrates good consistency. In contrast, the features selected by the ReliefF method exhibit significant spatial-distribution differences in the wheat stripe rust among the different monitoring results, indicating poor stability and consistency. Overall, incorporating information on disease spatial-distribution differences in stripe-rust monitoring can improve the accuracy and stability of disease monitoring, and it can provide data and methodological support for regional stripe-rust detection and accurate preventions.

**Keywords:** wheat stripe rust; geographical detectors; remote sensing; spatial-distribution difference; monitoring





## 1. Introduction

Wheat stripe rust is an airborne disease with a fast spreading speed and wide epidemiological range, occurring in more than 60 countries around the world [1]. It leads to an annual yield reduction of more than 5 million tons globally, and it indirectly causes economic losses of up to USD 1 billion [2,3]. China, in particular, is heavily affected by stripe rust, with an annual occurrence area of approximately 4 million hectares, which results in a wheat yield loss of over 1 million tons [4]. Given the severe damage caused by stripe rust, the timely and accurate detection of infected wheat fields and the implementation of control measures are of vital importance in safeguarding wheat yield, reducing pesticide usage, and minimizing economic losses [5]. The traditional monitoring of wheat stripe rust relies mainly on field surveys and visual discrimination, which has a high monitoring

accuracy but is limited to point-based observations. These traditional methods are prone to disease underreporting, leading to severe yield losses due to untimely control, and are inadequate in terms of meeting the demands of large-scale crop disease monitoring and precise control [6]. Remote sensing has been widely applied to crop and forest disease monitoring in recent years due to its ability to rapidly acquire continuous spatio-temporal observations over a wide area [7–9].

After being affected by diseases, crops undergo changes in their physiological, biochemical, and structural parameters, resulting in symptoms such as chlorosis and yellowing [10]. This leads to differences in the spectral reflectance between healthy and diseased crops [11]. By utilizing these differences, spectral features can be constructed for the remote sensing monitoring of crop diseases [12]. For instance, based on the spectral response mechanism of wheat stripe rust, Zheng et al. constructed the Red Edge Disease Stress Index (REDSI) by using the red edge and red bands of Sentinel-2 imagery for the remote sensing monitoring of regional wheat stripe rust [13]. Singh et al. used thermal infrared indices, such as the Crop Water Stress Index (CWSI) and the stomatal conductance index (IG), to characterize the changes in the respiration and evapotranspiration of wheat induced by stripe rust, and combined them with visible spectral signatures for wheat-stripe-rust monitoring [14]. In some scenarios when changes in crop spectral reflectance are caused by diseases that are prone to being confused with other factors, such as nitrogen–water stress and the growth stage, the spectral features may not be sufficient for mapping [15,16]. Certain researchers have aimed to enhance the accuracy of crop disease monitoring by integrating spectral features with meteorological and textural features. Guo et al. combined vegetation indices with textural features to enhance the monitoring accuracy of wheat stripe rust; they found that the textural features extracted at different spatial resolutions significantly influenced the monitoring accuracy [17]. Zheng et al. conducted regional wheat-stripe-rust monitoring by integrating meteorological data and Sentinel-2 data, demonstrating that the combination of meteorological and spectral features improved the monitoring accuracy of wheat stripe rust [18].

The progression of diseases from infection to symptom development is a continuous process, and certain researchers have attempted to improve monitoring and prediction accuracy by using multi-temporal remote sensing images within the disease development cycle. Based on the number of remote sensing images used, such studies can be classified into two categories: (1) dual-temporal and (2) multi-temporal. In the case of dual-temporal approaches, one remote sensing image is typically acquired before or early in the disease onset, while the other is obtained during the middle or late stages of disease development [19,20]. For example, Shi et al. constructed normalized two-stage vegetation indices using two PlanetScope images to map the damage from rice diseases. The research results demonstrated that the use of dual-temporal remote sensing images can mitigate the impact of phenological differences on rice disease monitoring [21]. Compared to dual-temporal approaches, multi-temporal remote sensing images provide more temporal information for disease monitoring [22]. Anderegg et al. utilized hyperspectral canopy data from 14 different dates to extract spectral–temporal features for the purpose of monitoring Septoria tritici blotch (STB). The results showed that the temporal changes in spectral reflectance could characterize the severity of STB, which is crucial for the efficient selection of resistant wheat varieties under field conditions [23]. Based on multi-temporal MODIS, Pryzant et al. utilized long short-term memory neural networks (LSTM) to extract the temporal features for predicting wheat stripe rust, thus successfully forecasting the occurrence of stripe rust in Ethiopia [24]. Based on multispectral data at multiple time points within the wheat-blast disease cycle, Gongora-Canul et al. quantified the severity of spike blast using a digital approach based on nongreen pixels and found that it was accurate and precise at moderately low to high visual wheat-spike-blast severity levels [25].

In addition to temporal information, spatial information such as heterogeneous spatial-distribution characteristics and farmland landscape features have also been widely used. For example, Zhao et al. found that spatial information is crucial in the classification of

heterogeneous crops, and the inclusion of spatial information can improve the performance of classification [26]. Liu et al. used a point process model to explore the spatial distribution patterns of pine wilt disease and quantitatively analyzed the response relationships between terrain factors, vegetation factors, human factors, and pine wilt disease [27]. Zhang conducted a landscape pattern analysis based on a land cover dataset and found that the more complex the land cover types and the larger the rice planting patch area, the higher the peak incidence of rice bacterial leaf blight was [28]. The occurrence and development of crop diseases and pests are the result of interactions between pathogens, hosts, and the environment [29]. Different periods of agricultural environmental conditions and crop growth statuses result in different spatial-distribution characteristics of diseases and pests within a region, which can be used for disease and pest monitoring. For example, Backoulou et al. first used supervised classification to distinguish between healthy and stressed wheat, and then combined spatial pattern indicators extracted from multispectral imagery, such as connectivity and the aggregation of patches, with topographic and soil variables to successfully distinguish wheat that was stressed by Russian wheat aphid from wheat that was stressed by other factors, such as drought and agronomic conditions [30].

The occurrence and prevalence of wheat stripe rust are influenced not only by pathogen and environmental conditions, but also by other factors such as wheat variety resistance, cropping systems, and topography [31–33]. These multiple factors contribute to a spatially heterogeneous distribution of wheat stripe rust at the regional scale. Existing studies have mostly focused on the fusion of multi-source data and the construction of key monitoring features to improve disease monitoring accuracy, while paying less attention to the regional distribution characteristics of the disease. In light of this, the specific objectives of this study are as follows: (1) Based on the developmental patterns of wheat stripe rust in the study area, integrate multi-source data including remote sensing, meteorological, and phenological data to construct monitoring features closely related to wheat stripe rust. (2) Utilize geographical detectors to select the optimal feature set for wheat-stripe-rust monitoring, considering the spatial distribution characteristics of wheat stripe rust in the study area. (3) Based on the final monitoring results, evaluate the effectiveness of geographical detectors for stripe-rust monitoring.

## 2. Materials and Methods

### 2.1. Field Survey and Data Collection

2.1.1. Study Area and Field-Experimental-Data Acquirement

Field surveys of wheat yellow rust were conducted in Qishan County, Baoji City, Shaanxi Province, China (34°18′–34°30′N, 107°33′–107°47′E) (Figure 1). The county, which is one of the major grain production counties in China, has a cultivated land area of 35.3 thousand hectares, with the main crop being winter wheat. It is adjacent to the over-wintering and over-summer regions of the Chinese wheat stripe rust, with abundant sources of the pathogen [4]. And the area is located in the western part of the Guanzhong Plain, which experiences a temperate semi-humid climate with an average annual temperature and precipitation of 6~13 °C and 500~700 mm, respectively [34]. The suitable climatic conditions coupled with sufficient fungal sources make the stripe-rust disease a frequent and severe occurrence in the area. According to the statistics from the Agro-Tech Extension Service Center in Qishan County, a severe infestation occurred in the study area in 2021, with significant differences in its occurrence levels among the different regions (http://jcnj.cbpt.cnki.net, accessed on 25 July 2022).

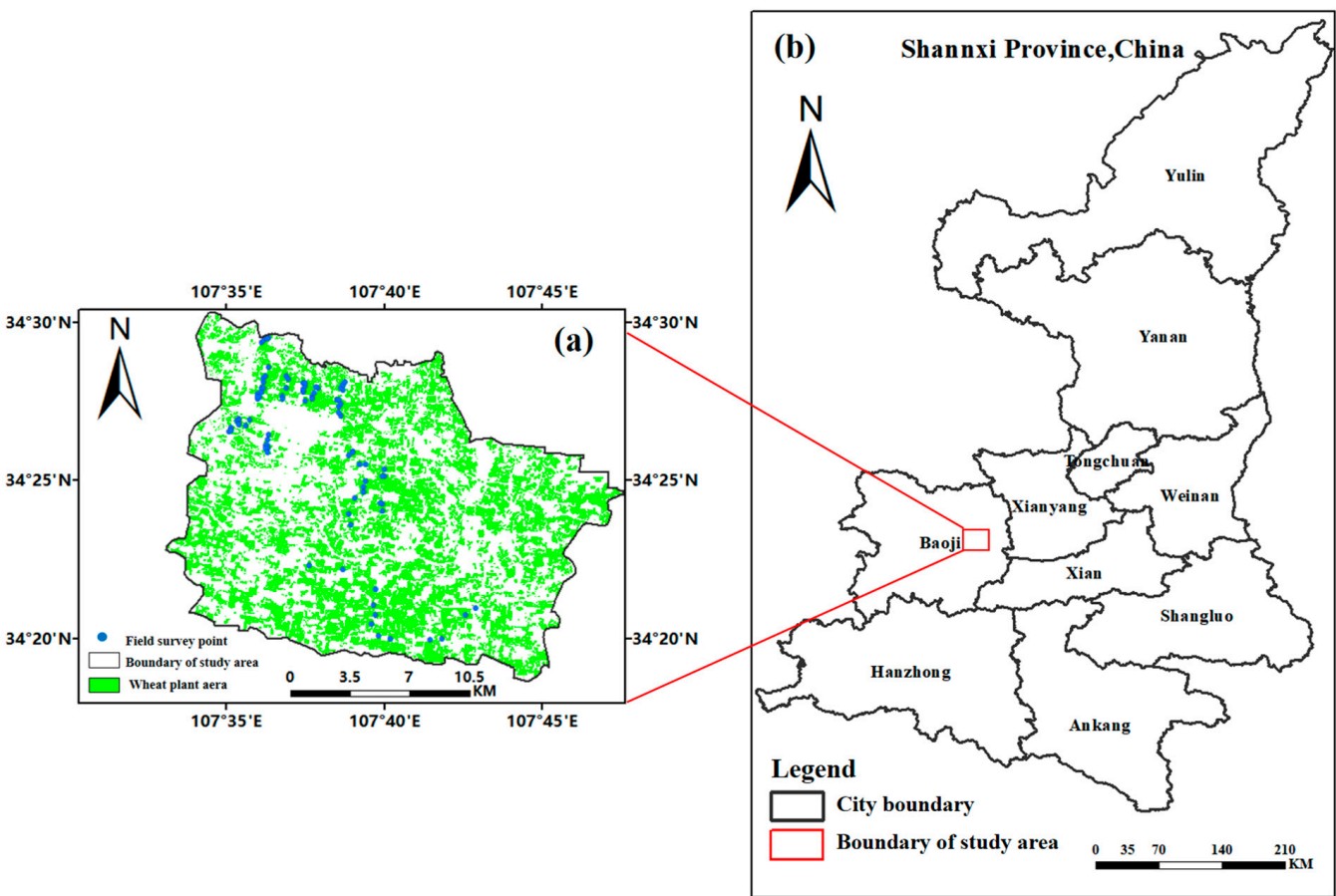

**Figure 1.** The geographical location of the study area and spatial distribution of sample points. (**a**) The distribution of wheat fields and survey points within the study area. (**b**) The approximate location of the study area in Baoji City, Shaanxi Province, China.

The field survey experiment was conducted from 28th to 30th April 2021. A total of 94 sample points were investigated, including the disease index (DI) of wheat stripe rust and the latitude and longitude coordinates of the sample points. The DI was measured according to the "National Rules for the Investigation and Forecasting of Crop Diseases" (GB/T 15795-2011) [35], and the specific calculation formula is shown in Equation (1):

$$DI = \alpha \times \theta \times 100, \theta = \frac{\sum(i \times m_i)}{M}, \alpha = \frac{m}{M} \tag{1}$$

where $\alpha$ is the rate of diseased leaves, $\theta$ is the average severity of infection at the sample point, $i$ is the severity level of the leaf, which is divided into nine categories (0%, 1%, 5%, 10%, 20%, 40%, 60%, 80%, and 100%), $m_i$ is the number of leaves of severity level $i$, $m$ is the total number of diseased leaves at the sample point, and $M$ is the total number of leaves observed at the sample point. To avoid the influence of mixed pixels on the monitoring results, the experiment selected a 10 m × 10 m area within a relatively uniform 20 m × 20 m wheat field as the survey sample point. The five-point sampling method was employed to determine the DI for each sample point, an area of 1 m × 1 m was selected at the center, four corners of each survey sample point were taken to record the DI, and then the average value was taken as the DI of the survey sample point. The coordinates of the sample points were measured using a handheld Global Positioning System (GPS) receiver.

2.1.2. Meteorological Data and Preprocessing

The meteorological data used in this study include daily meteorological data and agricultural meteorological data. The daily meteorological data consist of the average temperature (TEM), maximum temperature (HTEM), minimum temperature (LTEM), relative humidity (RHU), sunshine duration (SSD), precipitation (PRE), and average wind speed (WIN) recorded at the meteorological stations, which were downloaded from the China Meteorological Data Service Centre. Based on the occurrence patterns of wheat stripe rust in the study area, the daily meteorological data from surrounding meteorological stations for the period from January to June 2021 in the study area were obtained. To obtain continuous surface gridded daily meteorological data, we utilized a digital elevation model (DEM) as a covariate and employed a thin plate smoothing spline provided by ANUSPLINE software for spatial interpolation, with a resolution of 10 m [36]. The DEM used in the study was ASTER GDEM, downloaded from the Google Earth Engine (GEE). The agricultural meteorological data were the wheat phenological data recorded at agricultural meteorological stations, which provided detailed information about the time when most of the wheat in the vicinity of these stations reached specific growth stages (e.g., planting, heading, milk stage, etc.). We obtained wheat phenological data from the Fengxiang agricultural meteorological station ($34°31'$N, $107°23'$E) for the years 2019 to 2021. Based on this data, the wheat phenology in the study area were extracted and the accuracy was verified.

2.1.3. Remote Sensing Data and Preprocessing

Sentinel_2 satellite data with less than 20% cloud cover were acquired from the GEE, and these were surface-reflectance products that had undergone atmospheric and radiometric correction, as well as cloud detection. A total of 49 usable Sentinel-2 images were available for the study area from August 2020 to July 2021. Additionally, for the Fengxiang agricultural meteorological station, 46 images were available for the periods August 2019 to July 2020, as well as 43 images for August 2020 to July 2021. Each Sentinel-2 image underwent two preprocessing steps: cloud masking and resampling. Cloud removal was achieved by using the 'probability' band from the S2_CLOUD_PROBABILITY product, where pixels that had values greater than 65 were identified as clouds and subsequently removed; in addition, each band of the Sentinel-2 image was resampled to 10 m. Both steps were performed within the GEE platform. We employed a combination of decision trees and multi-temporal phenological information to extract the wheat planting area. The overall accuracy of the extraction results was verified using survey points and found to be 98% [37].

*2.2. Remote Sensing Monitoring of Wheat Stripe Rust at a Regional Scale Based on Geographical Detectors*

The remote sensing monitoring of wheat stripe rust at a regional scale based on geographical detectors mainly includes the following: (1) extracting key phenological stages of stripe-rust occurrence and development using Sentinel-2 time-series data; (2) the construction of monitoring features by integrating the multi-source data with phenological information, incorporating the spatial-difference information of the disease; and (3) employing geographical detectors to perform feature selection based on the spatial distribution of wheat stripe rust in the study area and establishing a monitoring model for disease detection. The overall research framework is illustrated in Figure 2.

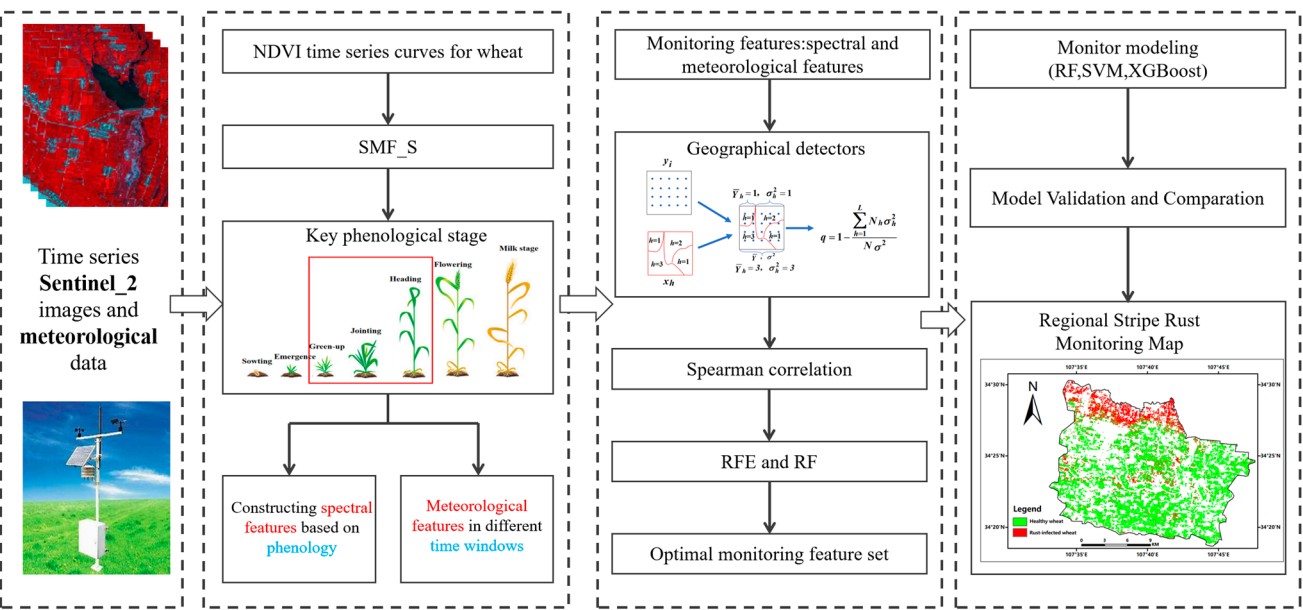

**Figure 2.** Methodological framework. The overall framework can be divided into three parts: (1) Construction of monitoring features based on phenological information. (2) Optimal selection of monitoring features using geographic detectors. (3) Monitoring model.

### 2.2.1. Extraction of Key Phenological Stages in Wheat

To address the limitations of the shape model fitting (SMF), through which it is difficult to describe the relationship between multiple phenological stages and to simulate the spatial and temporal variations of each phenological stage, Liu et al. developed shape model fitting via the separate phenological stage method (SMF-S) [38]. This method modifies the fitting function of SMF and employs an iterative approach within a locally adaptive window for shape matching. Compared to SMF, SMF-S provides a more reasonable spatial distribution of extraction results and lower extraction errors. Therefore, in this study, the SMF-S model was used to extract the key phenological stages of wheat in the study area. In the years of severe disease outbreaks in the study area, wheat stripe rust typically began to appear in the middle or late periods of green-up, and a large-scale epidemic was experienced during the heading period. Based on the occurrence and development patterns of wheat stripe rust in the study area, we extracted the key phenological stages of wheat, which include the green-up, jointing, and heading stages.

The process of wheat phenology extraction was as follows: (1) Reference curve construction: Sentinel_2 time-series images from 2019 to 2020 at the Fengxiang agricultural meteorological station were utilized to build an NDVI time-series curve. The Savitzky–Golay filtering algorithm that was improved by Chen et al. was applied to smooth the NDVI curve [39]. Based on the wheat phenological data recorded at the Fengxiang agricultural meteorological station from 2019 to 2020, the position of each phenological point on the smoothed NDVI curve was determined. (2) Accuracy validation and phenology extraction: This was based on the Sentinel_2 time-series images of the study area and Fengxiang agrometeorological station from 2020 to 2021. The phenology extraction was carried out by using the constructed reference curves and the SMF-S model, and the accuracy verification was then performed.

### 2.2.2. Multi-Source Feature Construction for Wheat-Stripe-Rust Monitoring

After being infected by wheat stripe rust, the chlorophyll content in wheat decreases, which will, thus, lead to a reduction in the photosynthetic rate. As the disease progresses, the leaf surface becomes damaged, transpiration increases, and the plant water content decreases, resulting in leaf curling [40]. Based on the mechanisms of stripe-rust infection,

we selected the vegetation indices related to wheat water content, chlorophyll content, canopy structure and coverage, and the stress status, for wheat-stripe-rust monitoring. The formulas for calculating these vegetation indices are shown in Table 1. Building upon the extracted wheat phenology, we computed the vegetation indices for wheat during the green-up, jointing, and heading stages for each pixel. If there were no available Sentinel_2 remote sensing images for a specific period, the nearest image was used as a replacement. Additionally, we calculated single-date vegetation indices via the Sentinel_2 image for the study area on 1 May 2021. Furthermore, we computed the normalized values and ratios of each vegetation index from the green-up to jointing stages and from the jointing to heading stages, as shown in Formulas (2) and (3):

$$\mathrm{VI_n} = \frac{\mathrm{VI_2} - \mathrm{VI_1}}{\mathrm{VI_2} + \mathrm{VI_1}} \tag{2}$$

$$\mathrm{VI_r} = \frac{\mathrm{VI_2}}{\mathrm{VI_1}} \tag{3}$$

where $\mathrm{VI_1}$ represents the vegetation index at the previous moment, and $\mathrm{VI_2}$ represents the vegetation index at the subsequent moment.

**Table 1.** Vegetation indices that are sensitive to wheat stripe rust.

| Correlation | Vegetation Indices | Formula |
|---|---|---|
| Water content | Moisture Stress index, MSI [41] | $R_{SWIR}/R_{NIR}$ |
| | Disease Water Stress Index, DSWI [42] | $(R_{NIR} + R_G)/(R_{SWIR} + R_R)$ |
| | Shortwave Infrared Water Stress Index, SIWSI [43] | $(R_{NIR} - R_{SWIR})/(R_{NIR} + R_{SWIR})$ |
| Pigment content | Green Leaf Index, GLI [44] | $(2R_G - R_R - R_B)/(2R_G + R_R + R_B)$ |
| | Greenness Ratio Vegetation index, GRVI [45] | $R_G/R_R$ |
| | Modified Chlorophyll-Absorption-Ratio Index, MCARI$n$ [46] | $((R_{ren} - R_R) - 0.2(R_{ren} - R_G)) * (R_{ren}/R_R)$ |
| | Red–Green–Blue Vegetation Index, RGBVI [47] | $(R_G^2 - R_B * R_R)/(R_G^2 + R_B * R_R)$ |
| | Structure-Independent Pigment Index, SIPI [48] | $(R_{NIR} - R_B)/(R_{NIR} - R_R)$ |
| | Normalized Difference Vegetation Index, NDVI [49] | $(R_{NIR} - R_R)/(R_{NIR} + R_R)$ |
| | Green-Normalized Difference Vegetation Index, GNDVI [50] | $(R_{NIR} - R_G)/(R_{NIR} + R_G)$ |
| | Excessive Green Index, ExG [51] | $2R_G - R_R - R_B$ |
| Vegetation coverage | Atmospherically Resistant Vegetation Index, ARVI [52] | $(R_{NIR} - 2R_R + R_B)/(R_{NIR} + 2R_R - R_B)$ |
| | Difference Vegetation Index, DVI [53] | $(R_{NIR} - R_R)$ |
| | Enhanced Vegetation Index, EVI [50] | $2.5(R_{NIR} - R_R)/(R_{NIR} + 6R_R - 7.5R_B + 1)$ |
| | Modified Simple Ratio Index, MSR [54] | $(R_{NIR}/R_R - 1)/(\sqrt{R_{NIR}/R_R} + 1)$ |
| | Optimized Soil-Adjusted Vegetation Index, OSAVI [55] | $(R_{NIR} - R_R)/(R_{NIR} + R_R + 0.16)$ |
| | Renormalized Difference Vegetation Index, RDVI [56] | $(R_{NIR} - R_R)/(\sqrt{R_{NIR} + R_R})$ |
| | Simple Ratio Index, SR [57] | $R_{NIR}/R_R$ |
| Stress status | Normalized Difference Vegetation Index Red Edge, NDVIrel$n$ [49] | $(R_{NIR} - R_{ren})/(R_{NIR} + R_{ren})$ |
| | Normalized Red-edge 1 Index, NREDI1 [58] | $(R_{re2} - R_{re1})/(R_{re2} + R_{re1})$ |
| | Normalized Red-edge 2 Index, NREDI2 [58] | $(R_{re3} - R_{re1})/(R_{re3} + R_{re1})$ |
| | Normalized Red-edge 3 Index, NREDI3 [58] | $(R_{re3} - R_{re2})/(R_{re3} + R_{re2})$ |
| | Plant Senescence Reflectance Index, PSRI$n$ [59] | $(R_R - R_G)/R_{ren}$ |
| | Red-edge Disease Stress Index, REDSI [13] | $((705 - 665)(R_{Re3} - R_R) - (783 - 665)(R_{Re1} - R_R))/(2R_R)$ |
| | Red-edge Inflection Point, REIP [60] | $705 + 35((R_R + R_{Re2})/2 - R_{NIR})/(R_{Re1} - R_{NIR})$ |
| | Triangular Vegetation Index, TVI [61] | $0.5(120(R_{NIR} - R_G) - 200(R_R - R_G))$ |
| Band | $R_B, R_G, R_R, R_{NIR}, R_{re1}, R_{re2}, R_{re3}, R_{SWIR}$ | |

Note: $R_B, R_G, R_R, R_{NIR}, R_{re1}, R_{re2}, R_{re3}, R_{SWIR}$ represents the bands B2, B3, B4, B5, B6, B7, and B11 of the Sentinel_2 imagery. $R_{ren}$ where $n$ = 1, 2, or 3, corresponds to the three red-edge bands of Sentinel_2 imagery, namely, B5, B6, B7.

The occurrence and development of crop diseases are closely related to the status of the hosts and the environmental factors within a certain time range. For instance, in the early stages of stripe rust spread, wheat fields with early planting in favorable environmental conditions are more susceptible to infection due to their close proximity. Therefore, in this study, we combined wheat phenology with environmental factors to extract the meteorological monitoring features that influence the occurrence and development of wheat stripe rust through a window analysis approach. We considered the green-up and jointing stages as

the midpoint of the window and the heading as the endpoint; furthermore, we calculated the average values of the meteorological features within the windows of lengths of 7, 15, and 21 days, respectively. These features included the maximum temperature, minimum temperature, average temperature, sunshine duration, relative humidity, precipitation, and average wind speed. To investigate the relationship between long-term meteorological variables and wheat stripe rust during different phenological stages, we also calculated the average values of the aforementioned meteorological features from the green-up to the jointing stages and from the jointing to heading stages.

2.2.3. Feature Selection for Disease and Pest Monitoring Using Geographical Detectors

After constructing the multi-source features for stripe rust monitoring, the next step is to select appropriate monitoring features for modeling. Due to the spatial variation in the distribution of wheat stripe rust within the study area, geographical detectors were introduced for feature selection. Geographical detectors are valuable tools for exploring spatial-data analysis, and they are primarily used to detect spatial differentiation and to reveal the underlying driving forces. The core idea is that if a certain independent variable has a significant impact on a dependent variable, their spatial distributions should exhibit similarity [62]. In this study, we mainly utilized factor detection for the geographical detectors. The factor detection involved the use of the *q-value* to indicate the explanatory power of the independent variable *X* on the spatial distribution of the dependent variable *Y*, expressed as follows:

$$q = 1 - \frac{\sum\limits_{h=1}^{L} N_h \sigma_h^2}{N \sigma^2} \tag{4}$$

where $h$ is the number of strata or categories of the dependent variable $X$, and $N_h$ and $N$ are the numbers of units in stratum $h$ and the entire population, respectively. $\sigma_h^2$ and $\sigma^2$ are the variances of the samples within stratum $h$ and the entire population, respectively. The q-value ranges from 0 to 1, where a higher value indicates a stronger similarity between the spatial distributions of the independent $X$ and dependent variables $Y$, indicating the greater importance of the independent variable $X$. It should be noted that factor detection is suitable for handling categorical variables, while the features constructed in this study are continuous variables. To address this, we utilized the Optimal Parameters-Based Geographical Detector Model (OPGD) proposed by Song et al. to discretize the continuous data [63].

Considering that redundant features can reduce model monitoring accuracy and increase computational complexity, this study employed the Spearman correlation coefficient to remove redundant features. If the correlation between two features was greater than 0.8, then the feature with the lower q-value was removed, while the feature with the higher q-value was retained. However, certain features may have low q-values, but when combined with other features they significantly improve the model's monitoring accuracy. Using a threshold-based method alone to select features may not yield the optimal monitoring feature set. Therefore, this study utilized a combination of recursive feature elimination (RFE) and RF methods to achieve the optimal feature selection. RFE is a backward selection method that iteratively trains the model and removes features to gradually reduce the size of the feature set until the best feature subset is selected [64]. The combination of RFE and RF provides a powerful approach for feature selection, helping to identify the most informative features for the monitoring model [65].

In order to validate the effectiveness of the geographical detectors in wheat stripe rust monitoring, this study selected the commonly used ReliefF feature selection algorithm for comparison. In recent years, the ReliefF algorithm has been increasingly adopted by scholars in the remote sensing field and applied to various related research studies [66–68]. The core idea of ReliefF is to assess the discriminative ability of each feature variable with respect to the k-nearest neighbor samples [69]. It increases the weight of features that contribute to distinguishing samples of different classes while reducing the weight of

features that have a negative impact on distinguishing different class samples. In the end, ReliefF provides weights for each feature based on its discriminative power.

### 2.2.4. Monitoring Modeling

Considering that in the early stages of stripe-rust infection, there is little spectral difference between healthy and diseased wheat at the canopy scale, to improve the reliability and accuracy of monitoring, we classified survey points with DI $\leq$ 10 as healthy samples, and those with DI > 10 as diseased samples. This transformed the modeling problem into a binary classification task. Due to the limited number of survey points, 10-fold cross-validation was employed to train and validate the models. The overall accuracy (OA) and Kappa coefficient were used to evaluate the monitoring accuracy of the models. We selected the Random Forest (RF), eXtreme Gradient Boosting (XGBoost), and Support Vector (SVM) models for stripe-rust monitoring modeling.

RF has been widely used in agricultural remote sensing, such as crop area extraction, wheat yield estimation, and crop pest and disease monitoring [70–72]. RF is an ensemble learning model that constructs a strong learner by combining multiple independent decision trees [73]. Previous results have shown that RF can achieve superior generalization ability in small-sample training, which is suitable for our research [74]. XGBoost is also an ensemble learning model, but it differs from RF in that it trains multiple decision trees in a serial manner, with each tree being based on the predictions of the previous one. Compared to RF, its ensemble method is more efficient [75]. SVM is one of the most commonly used machine learning algorithms in vegetation pest and disease monitoring [22]. It predicts input samples based on a small number of support vectors, and the large number of redundant samples in the training set does not affect the prediction results. This model has good robustness and is particularly suitable for learning classification in small samples [76]. Previous research has demonstrated that these three classification algorithms can be used for disease monitoring and have high monitoring accuracy [16,20]. Furthermore, their model operating principles differ, making them suitable for testing the generalization ability of the feature set. XGBoost and SVM, in particular, were used to test the generalization ability of the feature set.

## 3. Results

### 3.1. Wheat Key-Phenological-Stage Extraction Results

The accuracy of the extracted key phenological stages of wheat was validated using the phenological data from the Fengxiang agricultural meteorological station for the years 2020–2021, as shown in Figure 3. For the green-up stage, the field observation date was February 3rd, and most of the wheat in the extraction results were located on 4th February and 12th February. As for the jointing stage, the field observations were made on 26th March, and the extraction results showed that 75% of the wheat were in the jointing stage between 24th March and 1st April. Regarding the heading stage, the field observation date was 23rd April, and the extraction results indicated that the majority of the wheat reached the heading stage on 24th April. The phenological extraction results demonstrated that during the early growth stages, there were significant differences in the growth status among the different wheat plots. However, as the wheat continued to grow, the growth status gradually became more consistent, which is consistent with previous research findings [77]. Through validation, the extracted dates for various wheat phenological stages were found to be highly consistent with the dates recorded at the agricultural meteorological station. Based on the SMF-S model and the established reference curve, we extracted the key phenological stages of wheat in the study area, as shown in Figure 4. The extraction results indicated that the northwest and central regions of the study area exhibited significantly earlier jointing and heading dates compared to other regions, which aligns with the findings from the field surveys.

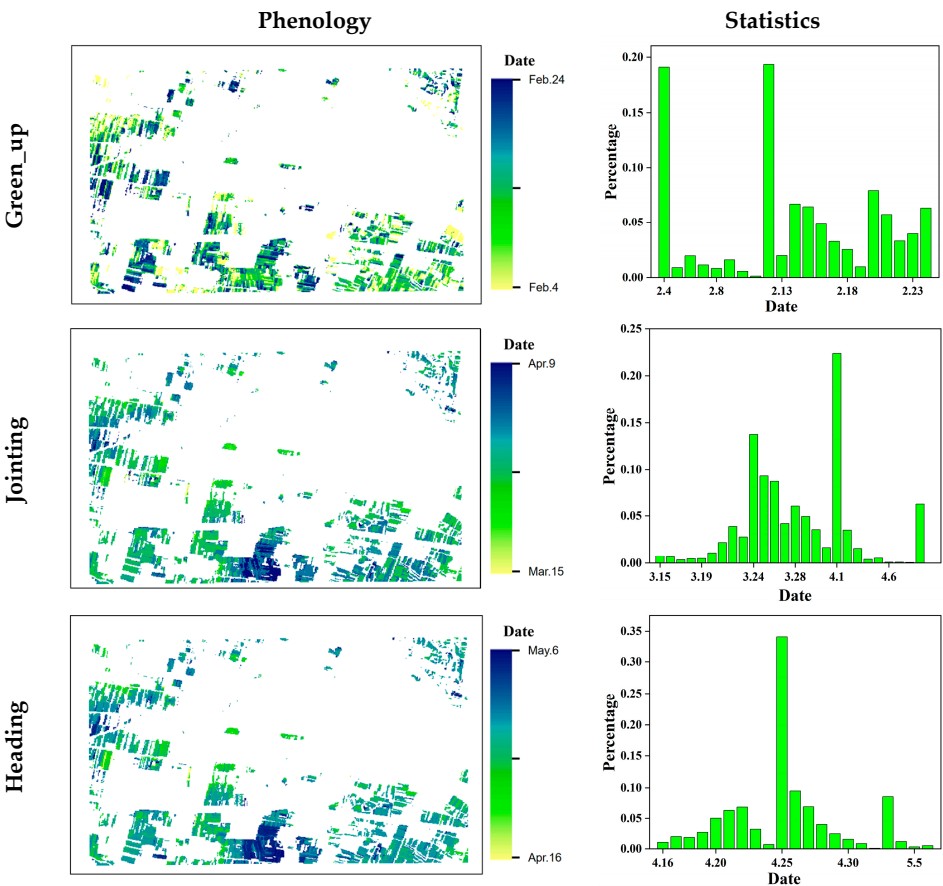

**Figure 3.** (**Left**) column shows the extracted results of the wheat phenological stages around the Fengxiang agricultural meteorological station. The (**right**) column displays a statistical analysis of the pixel count in each extraction result. The x-axis represents time, while the y-axis represents the proportion of pixels corresponding to that time in the entire extraction result.

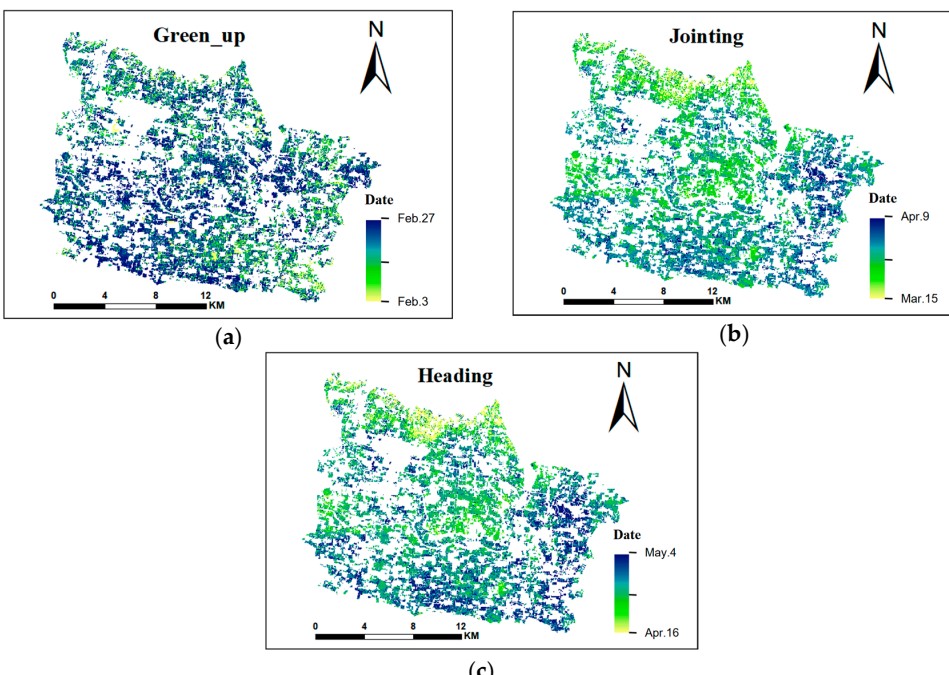

**Figure 4.** Results of the wheat-phenological-stage extraction in the study area. (**a–c**) represent the extraction results for wheat green-up stage, jointing stage, and heading stage in the study area, respectively.

*3.2. Feature Importance Analysis*

In this study, we utilized the geographical detectors and ReliefF methods to calculate the importance of the various features, and the top 20 ranked features are listed in Figure 5. The results from the geographical detectors method revealed a strong correlation between the temperature and the occurrence and development of wheat stripe rust. Among the top 10 ranked features, 9 were related to temperature. Furthermore, among the top 20 ranked features, 13 were temperature-related. Besides temperature, the sunshine duration before heading also exhibited significant importance. In the ReliefF calculation results, the top 10 ranked features included precipitation, sunshine duration, relative humidity, and temperature, with precipitation and temperature having higher importance. Among the top 20 ranked features, 7 were temperature-related, and 5 were precipitation-related. Both the geographical detectors and ReliefF results consistently demonstrated that meteorological features hold a greater weight when compared to spectral features. HTEM_H21, TEM_H21, LTEM_H21, LTEM_J15, LTEM_H15, SSD_H21, LTEM_H7, RHU_H21, WIN_H21, and PRE_JH are features that appeared in the results of both methods. This indicates that the pre-heading meteorological conditions, particularly temperature, significantly influence the development of wheat stripe rust in the study area.

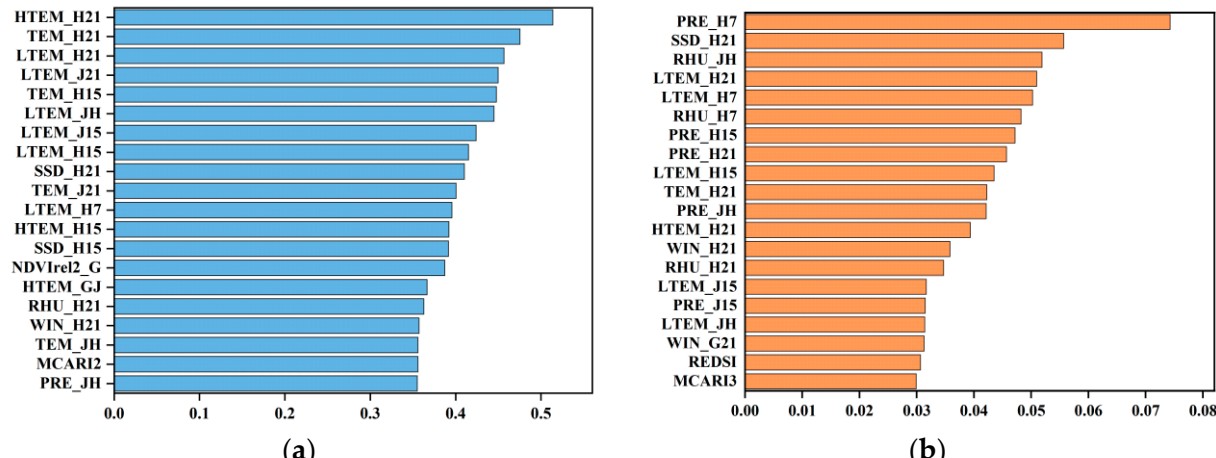

**Figure 5.** Top 20 ranked features in the geographical detectors' and ReliefF's calculation results. (1) Naming rules for the spectral features: xxx_t, xxx represents vegetation index, t represents the initial letter of the phenology, and MSR_H corresponds to the modified simple ratio (MSR) for wheat on the date of heading. It is worth noting that the two-stage normalized vegetation indices and two-stage ratio vegetation indices are named nxxx_t and rxxx_t, respectively. For example, nMSR_JH represents the normalized value of MSR from the jointing to heading stages. (2) Naming rules for the meteorological features: xxx_tm, xxx represents the meteorological variable, t represents the initial letter of the phenology, and m represents the window size or the initial letter of the phenology, e.g., SSD_H15 represents the average sunshine duration in the 15 days before the date of heading. (**a**) Geographical detectors' results; and (**b**) ReliefF results.

In order to further explore the differences between the top-ranked features in the Geographical detector and ReliefF calculation results, we performed normalization on the top 20 features from both methods. Subsequently, we calculated the mean differences and F-statistic between the healthy and diseased samples. The F-statistic expressed the significance of the differences between the healthy and diseased samples, which is calculated as the ratio of the variance between the groups to the variance within groups. A higher F-statistic value indicates a more significant difference between the healthy and diseased samples. Regarding the mean differences among the top 10 ranked features, except for the 2nd and 6th features (where the geographical detectors' mean differences were significantly larger than ReliefF), the mean differences for the other features were comparable between the two methods. As for the F-statistic, among the top 10 ranked features (with

the exception of the 4th and 9th features), the geographical detectors exhibited much larger F-statistic values for the remaining 8 features when compared to ReliefF. From Figure 6, it can be observed that in the last 10 features, for the F-statistic values, ReliefF produced mostly larger values than the geographical detector method. The reason for this is that the top 10 features in the geographical detectors' results were ranked from 10th to 20th in the ReliefF results. The results of the mean differences and F-statistic demonstrate that the top-ranked features selected by the geographical detector method, such as HTEM_H21, LTEM_H21, SSD_H21, etc., have better discriminative capabilities between the healthy and diseased samples.

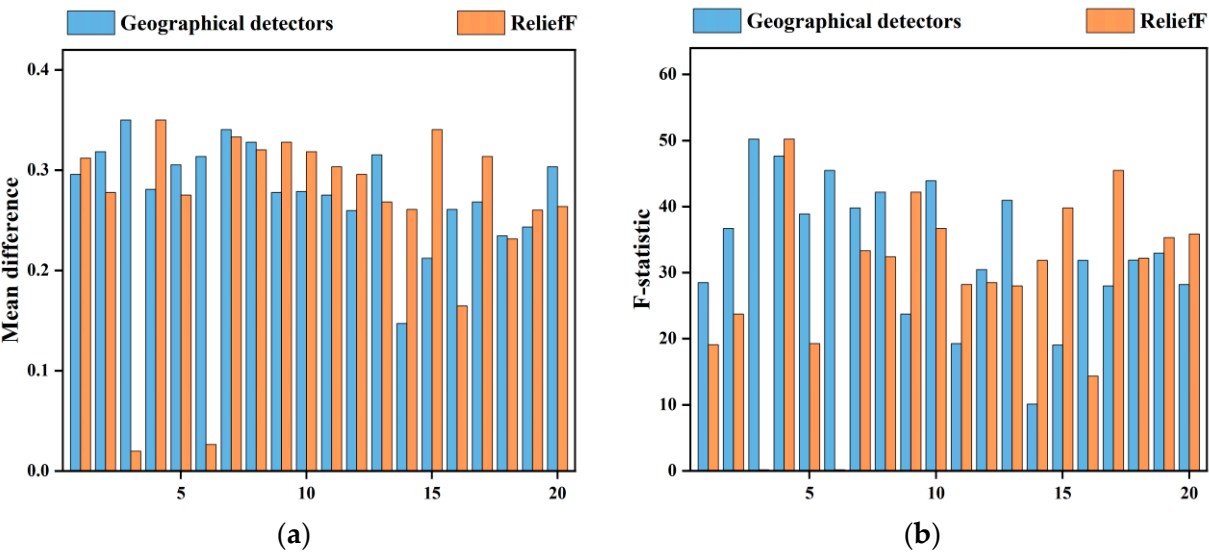

**Figure 6.** The mean differences and F-statistic between the healthy and diseased samples for the top 20 ranked features computed using geographic detectors and ReliefF, respectively. (**a**) Mean differences between the healthy and diseased samples. (**b**) F-statistic between the healthy and diseased samples.

### 3.3. Monitoring Feature Selection

After removing the redundant features, as based on feature importance and the correlation between features, we obtained the optimal monitoring feature set by using a combination of RFE and RF, named RFE_RF. To prevent overfitting caused by having too many features, we set a maximum limit of 30 features in the feature set. The final results are presented in Table 2 and Figure 7. The optimal monitoring feature set obtained through the combination of the geographical detectors and RFE_RF consisted of 11 features, while the optimal monitoring feature set obtained through the ReliefF and RFE_RF combination consisted of 6 features. The above two methods were named GD_RFE_RF and R_RFE_RF, respectively. The optimal feature sets obtained by GD_RFE_RF and R_RFE_RF were both composed of meteorological features and spectral features. It is noteworthy that the spectral features in both optimal monitoring feature sets represented wheat pigment content and stress status, e.g., SIPI and MCARI2 were used to characterize the wheat chlorophyll content, while NREDI2 and REDSI were used to represent the wheat stress status. Regarding the meteorological features, there were significant differences between GD_RFE_RF and R_RFE_RF. The former set included features like wind speed, relative humidity, temperature, and rainfall, while the latter set included features like sunshine duration, temperature, and rainfall. Notably, HTEM_J21, PRE_GJ, and SIPI were present in both optimal feature sets. Figure 7 shows that, compared to R_RFE_RF, most of the feature sets obtained using GD_RFE_RF with different numbers of features exhibited a higher accuracy.

**Table 2.** The optimal monitoring feature sets obtained through GD_RFE_RF and R_RFE_RF.

| Method | Number | Feature |
|---|---|---|
| Geographical detectors | 11 | HTEM_H21, LTEM_H7, HTEM_GJ, HTEM_J21, PRE_GJ, RHU_H15, WIN_J7, WIN_G21, NREDI2, SIPI, MCARI2 |
| ReliefF | 6 | SSD_H21, PRE_H15, HTEM_J21, PRE_GJ, SIPI, REDSI |

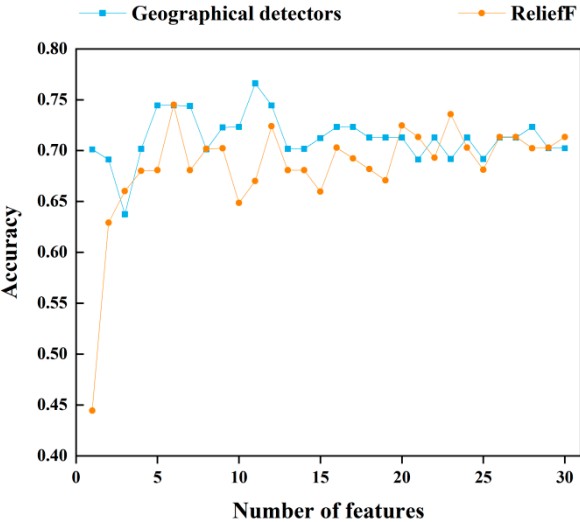

**Figure 7.** The monitoring accuracy of feature sets at different numbers of features obtained using GD_RFE_RF and R_RFE_RF. For GD_RFE_RF, the highest monitoring accuracy for the RF model was achieved when the feature set contained 11 features. For R_RFE_RF, the RF model exhibited the highest monitoring accuracy when the feature set contained 6 features.

Although the geographical detector method is based on the spatial distribution of the independent variable X and the dependent variable Y for model computation, its resultant value, the q-value, is a global value and cannot depict the spatial-distribution details of each independent variable. To clarify the spatial relationships between the features in the optimal monitoring feature set and DI, this study employed the multiscale geographically weighted regression (MGWR) method for further exploration. MGWR allows for the weighting and modeling of observation points at different spatial scales. During the MGWR modeling process, we standardized each feature to eliminate the dimensional differences between the different features. Figure 8 illustrates the regression coefficients of the 11 features at all sample points.

The constant term, also known as the intercept, represents the average level of the dependent variable's predicted values. In this study, it was correlated with the average DI, where higher values indicated more severe disease in the sampled points. The spatial distribution of the intercept revealed a gradual reduction in the severity of wheat stripe rust from the north to south. After standardization, the absolute values of the coefficients of the independent variables can be used to evaluate their impact on the dependent variable. Larger absolute values indicate a greater importance in the variable. Based on the absolute values of the regression coefficients, the features can be categorized into four levels as follows: (1) 0~0.1: LTEM_H7, PRE_GJ, RHU_H15, MCARI2—these features had the smallest importance. (2) 0.1~0.2: HTEM_H21, HTEM_GJ, NREDI2—these features had a relatively smaller importance. (3) 0.2~0.3: WIN_J7, WIN_G21—these features had a relatively higher importance. (4) 0.3~0.4: HTEM_J21, SIPI—these features had the highest importance. The spatial distribution of the regression coefficients of all the features showed that HTEM_H21, HTEM_J21, SIPI, and WIN_J7 exhibited significant differences in the coefficients across different regions. For example, the spatial distribution of the regression coefficient for HTEM_H21 indicated that the average maximum temperature in the 21 days before heading had a greater impact on the occurrence and development of wheat stripe

rust in the northern region compared to the southern and central regions. Apart from the above four features, the regression coefficients of the other features did not exhibit significant spatial-distribution differences across the entire study area.

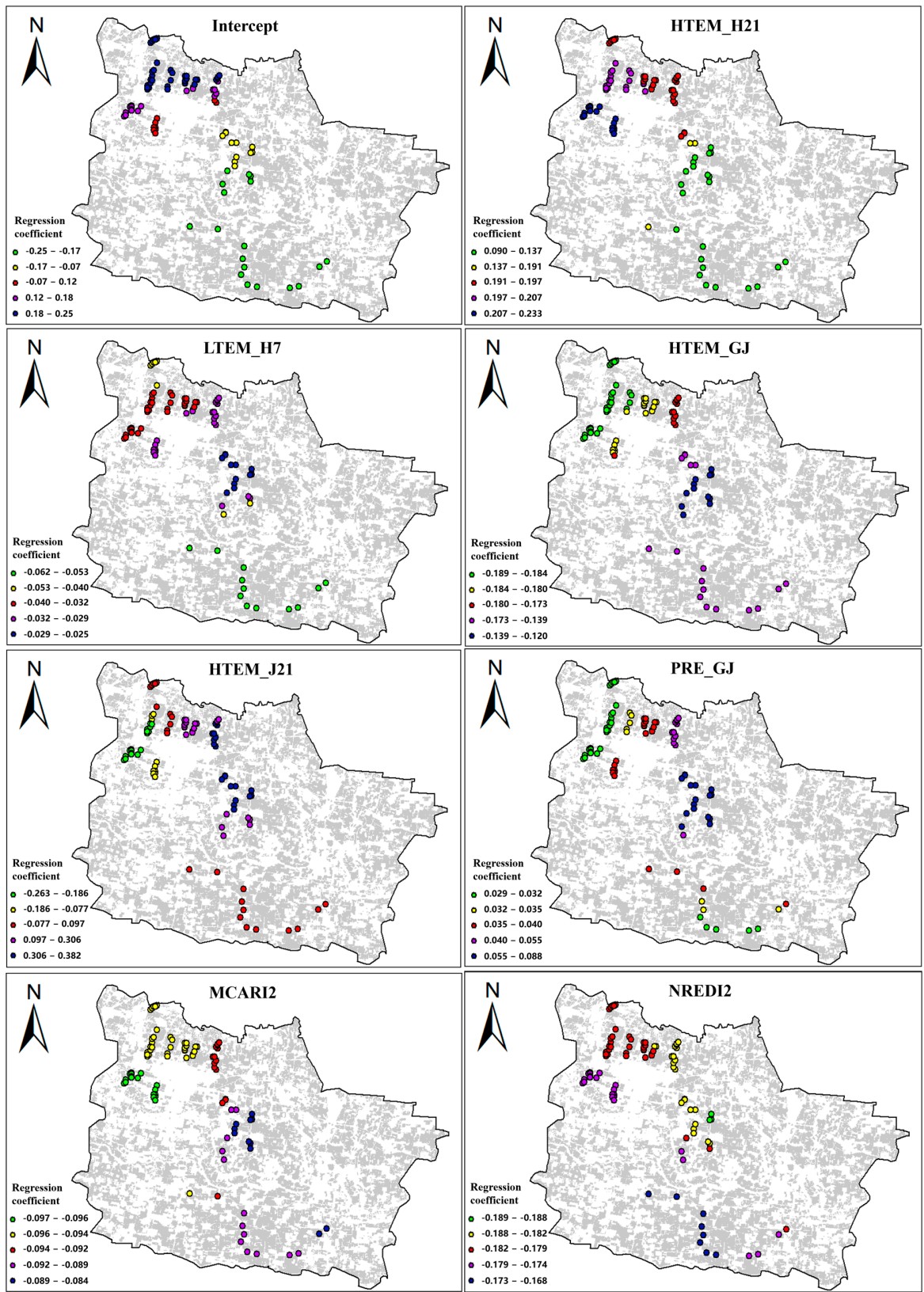

**Figure 8.** *Cont.*

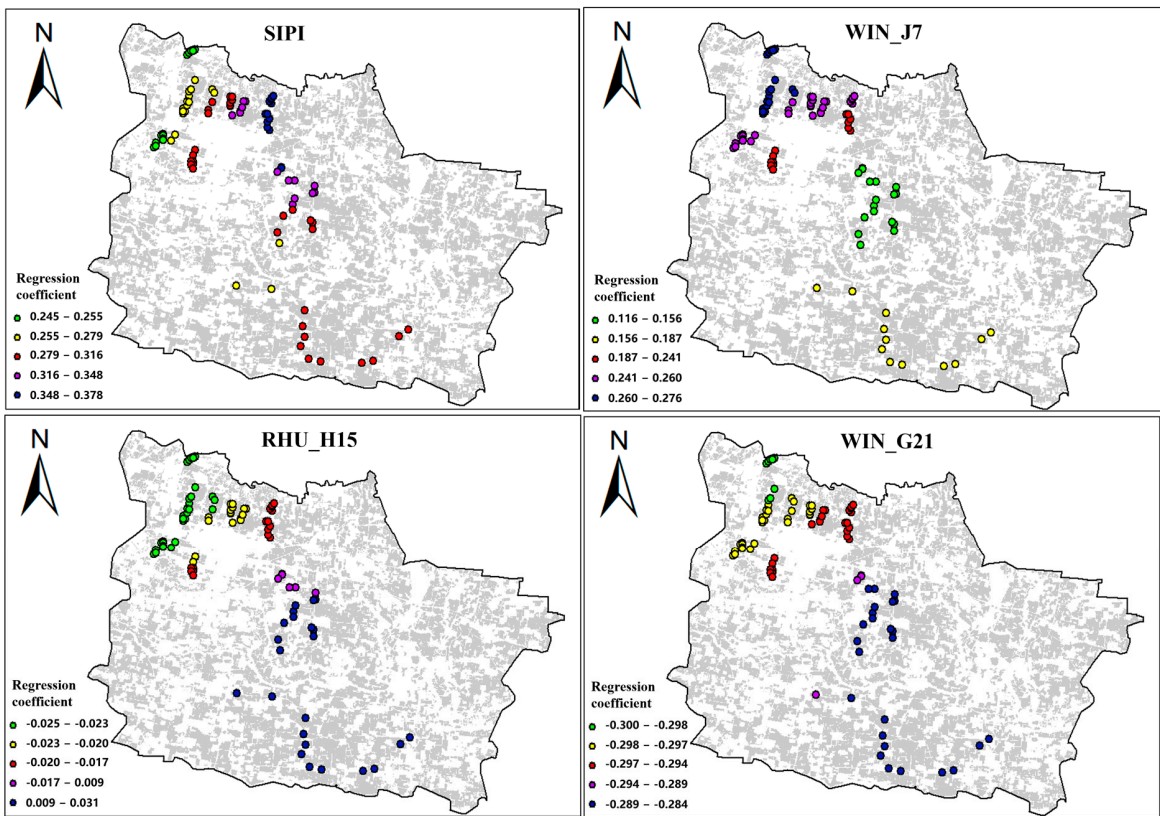

**Figure 8.** Spatial-distribution maps of the regression coefficients for each feature in the optimal monitoring feature set obtained using GD_RFE_RF.

### 3.4. Accuracy Validation of Monitoring Results

After obtaining the optimal monitoring feature set, we further optimized the RF monitoring model by using a grid parameter search. XGBoost and SVM were primarily used to test the stability of the optimal monitoring feature set. Six monitoring models were constructed, namely GD_RF, GD_XGBoost, GD_SVM, R_RF, R_XGBoost, and R_SVM. The optimal parameters for each monitoring model and the confusion matrices for the monitoring results are presented in Tables 3 and 4, respectively. The monitoring results show that GD_RF achieved the highest overall accuracy and Kappa coefficient, with 87.2% and 0.743, respectively. Compared to R_RF, it exhibited an improvement of 3.2% in its overall accuracy and 0.064 in its Kappa coefficient. When applying the optimal monitoring feature sets to XGBoost and SVM, there was a certain degree of decline in the monitoring accuracy. However, the optimal feature set obtained using the geographical detector method still demonstrated the highest monitoring accuracy. For XGBoost, GD_XGBoost achieved an overall accuracy of 80.9% and a Kappa coefficient of 0.614, which were 2.7% and 0.044 higher, respectively, than R_XGBoost. For SVM, GD_SVM achieved an overall accuracy of 74.5% and a Kappa coefficient of 0.484, which were 4.3% and 0.087 higher, respectively, than R_SVM. The model monitoring results indicate that the consideration of the spatial distribution characteristics of the disease can enhance the monitoring accuracy of wheat stripe rust.

**Table 3.** Optimal parameter settings for each monitoring model.

| Method | Parameters of RF | | Parameters of XGBoost | | Parameters of SVM | |
|---|---|---|---|---|---|---|
| | n_Estimators | max_Depth | n_Estimators | max_Depth | C | Gamma |
| Geographic Detector | 23 | 5 | 14 | 2 | 2 | 0.05 |
| ReliefF | 15 | 3 | 11 | 3 | 1 | 0.05 |

**Table 4.** Confusion matrix of the monitoring results of each model.

| Method | Model | | Healthy | Infected | Sum | UA | OA | Kappa |
|---|---|---|---|---|---|---|---|---|
| Geographic Detectors | RF | Healthy | 45 | 5 | 50 | 90.0% | | |
| | | Infected | 7 | 37 | 44 | 84.9% | 87.2% | 0.743 |
| | | Sum | 52 | 42 | 94 | | | |
| | | PA | 86.5% | 88.1% | | | | |
| | XGBoost | Healthy | 40 | 10 | 50 | 80.0% | | |
| | | Infected | 8 | 36 | 44 | 81.8% | 80.9% | 0.614 |
| | | Sum | 48 | 46 | 94 | | | |
| | | PA | 83.3% | 78.3% | | | | |
| | SVM | Healthy | 40 | 10 | 50 | 80.0% | | |
| | | Infected | 14 | 30 | 44 | 68.2% | 74.5% | 0.484 |
| | | Sum | 54 | 40 | 94 | | | |
| | | PA | 74.1% | 75.0% | | | | |
| ReliefF | RF | Healthy | 43 | 7 | 50 | 86.0% | | |
| | | Infected | 8 | 36 | 44 | 81.8% | 84.0% | 0.679 |
| | | Sum | 51 | 43 | 94 | | | |
| | | PA | 84.3% | 83.7% | | | | |
| | XGBoost | Healthy | 42 | 8 | 50 | 84.0% | | |
| | | Infected | 12 | 32 | 44 | 72.7% | 78.7% | 0.570 |
| | | Sum | 54 | 40 | 94 | | | |
| | | PA | 77.8% | 80.0% | | | | |
| | SVM | Healthy | 39 | 11 | 50 | 78.0% | | |
| | | Infected | 17 | 27 | 44 | 61.4% | 70.2% | 0.397 |
| | | Sum | 56 | 38 | 94 | | | |
| | | PA | 69.6% | 71.1% | | | | |

Note: OA, PA, and UA represent overall accuracy, producer accuracy, and user accuracy, respectively.

Based on the above monitoring models, we mapped the occurrence of the wheat-stripe-rust area across the entire study area, as shown in Figure 9. It can be observed that, except for R_SVM, the monitoring results of all other models exhibited the same spatial-distribution trend: the most severe disease occurrence was in the northwest wheat fields, while the central fields showed relatively mild occurrences, and the southern and other regions had the lowest occurrence rates. Further analysis of the spatial distribution details in the GD_RF and R_RF monitoring results revealed that the distribution of wheat stripe rust in GD_RF was more concentrated. For instance, in the central region, wheat stripe rust was mainly concentrated along the riverbanks, with particularly few occurrences in other fields. On the other hand, in the R_RF monitoring results, aside from the riverbanks, wheat stripe rust appears in patches in other fields, rather than being concentrated in specific areas. As the study area is one of China's major grain-producing counties, the timely

prevention and control of wheat stripe rust is a key focus of the local government's work. The government implements various measures, such as spraying pesticides in advance, to prevent the large-scale spread of stripe rust (http://nyncj.baoji.gov.cn/, accessed on 11 September 2022). Compared to R_RF, GD_RF's monitoring results are more consistent with the actual situation. When applying the two optimal monitoring feature sets to XGBoost and SVM, the feature set derived from the geographical detector model still showed a higher stability, with no significant spatial-distribution differences in the three models' monitoring results regarding wheat-stripe-rust occurrences. However, when applying the feature set derived from ReliefF to SVM, substantial discrepancies were observed when compared to XGBoost and RF. These results demonstrate that the feature set extracted based on the geographical detector model exhibits a higher stability compared to ReliefF.

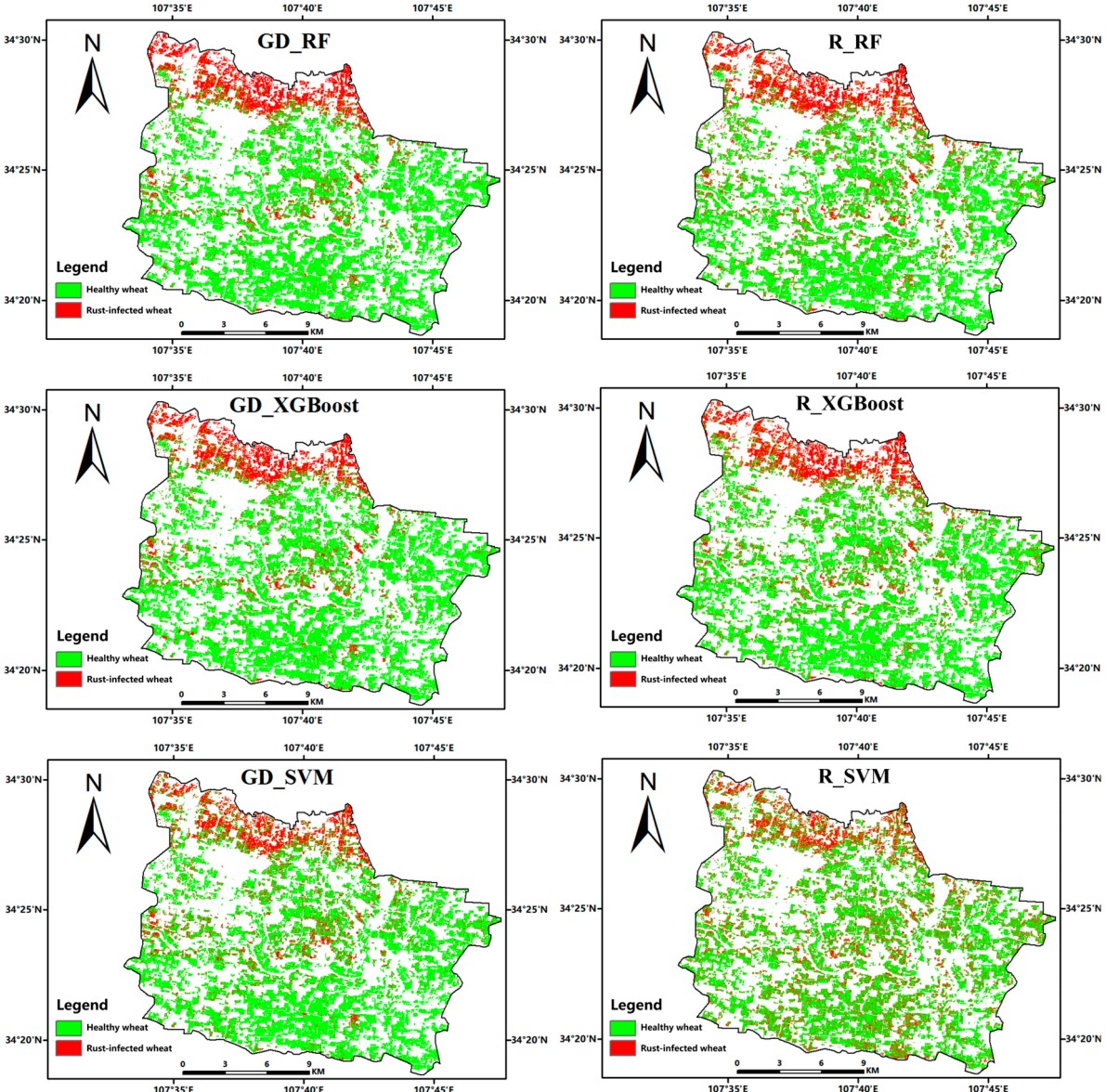

**Figure 9.** Mapping wheat yellow rust in the study area based on each monitoring model.

## 4. Discussion

### 4.1. Analysis of Spatial-Distribution Differences of Wheat Stripe Rust in the Study Area

According to the theory of disease triangle, the occurrence and epidemiology of wheat stripe rust is the result of the interaction of fungal sources, hosts and habitat conditions [78].

The spores of the rust fungus are transmitted over long distances by the wind, and rely on their own gravity or rainfall for landing [79]. Figure 10 shows that the northwestern wheat fields are close to the foothills; the complex terrain can provide more obstruction during spore transport. Compared to the flat central and southern regions, spores are more likely to accumulate in the northwestern region, providing ample pathogens for the occurrence of stripe rust. At the same time, the intricate terrain also increases the difficulty of wheat-stripe-rust prevention and control, leading to potential gaps in rust control measures. Previous studies have found a close relationship between the timing of crop planting and the severity of disease occurrence [80]. For example, Pan found that the earlier winter wheat is planted, the earlier stripe rust occurs and the more severe the disease becomes [81]. As indicated by the results of wheat phenology extraction in Section 3.1, wheat planting generally occurs earlier in the northwestern region. The early enclosure of wheat fields in this region creates a closed and conducive environment for the early spread of wheat stripe rust. In addition, the above-average rainfall in the first half of 2021 compared to previous years further accelerated the spread of stripe rust. These multiple factors have led to a higher incidence of wheat stripe rust in the northwestern region compared to other areas. Additionally, in the central region, wheat stripe rust mainly occurred in fields located near riverbanks. These areas received more irrigation due to their proximity to rivers, resulting in higher soil moisture levels. The moist field environment facilitates the infection and transmission of wheat stripe rust [82].

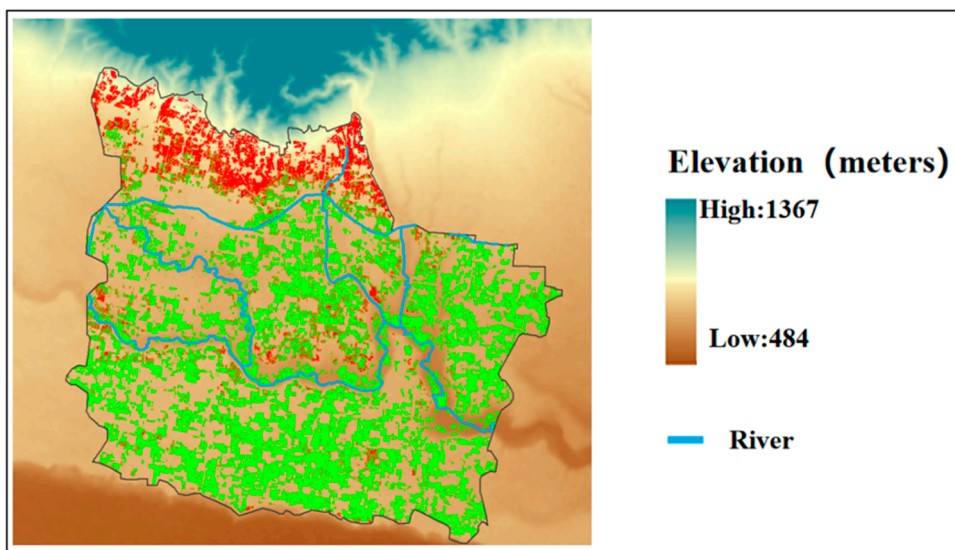

**Figure 10.** Elevation and river distribution map of the study area.

### 4.2. Analysis of the Performance of Spectral Features and Meteorological Features

After infection with stripe rust, physiological and biochemical parameters of wheat undergo changes, leading to differences in canopy spectra between healthy and infected wheat [11]. The current main research direction focuses on selecting or constructing key vegetation indices for stripe-rust monitoring based on the spectral response mechanism of infected wheat [12]. For example, Ruan et al. used a combination of ReliefF and sequential forward selection to obtain the optimal feature set for stripe-rust monitoring at different growth stages [35]. Their research results showed that at the heading stage, NDVI for representing wheat pigment content, DSWI for representing wheat water content, and REDSI, TVI, PSRIre2, NDVIre2 for representing wheat stress status were the optimal monitoring feature set. In this study, only SIPI, MCARI2, and NREDI2 were used, and spectral features representing wheat water content were not included. In the study area, farmers select wheat varieties under the guidance of local governments, primarily including XiNong822, BaiNong207, and others. These wheat varieties exhibit a certain level of resistance to stripe rust while maintaining relatively high yields. Wu et al. found that disease-resistant wheat

showed a slow decline in its relative water content after infection with stripe rust, followed by a recovery, thus leading to a lower reduction in relative water content compared to healthy wheat [83]. Additionally, the study did not incorporate spectral features representing wheat canopy structure. Wheat typically exhibits leaf curling symptoms only in the later stages of infection, leading to a reduction in canopy cover [40]. However, the most infected wheat was at the early and middle stages of disease development. These factors contribute to the diminished significance of spectral features used to characterize the water content and the canopy structure of wheat.

The propagation, infestation, and reproduction of stripe rust require suitable environmental conditions; for example, high-humidity and low-temperature environments are more favorable for spore infestation and reproduction [84]. Figure 8 demonstrates the close correlation between wind speeds (WIN_J7, WIN_G21) during the green-up and jointing stages and the development of wheat stripe rust. This is mainly attributed to the study area being located in a spring epidemic region of wheat stripe rust in China, where the severity of the disease depends on the timing and quantity of spore introduction in the spring [4]. Wind serves as a crucial carrier for spore dissemination, controlling the spread distance and density of spores, thus influencing the early spread of wheat stripe rust. Temperature and humidity are key factors influencing the occurrence and development of wheat stripe rust [85]. Spore infection requires dew, and higher relative humidity leads to increased dew formation, subsequently elevating the incidence of stripe rust [80,86]. Temperature mainly affects the efficiency of spore infection. The MGWR regression coefficients showed that temperature (HTEM_J21 and HTEM_H21) had a relatively higher importance. However, humidity (RHU_H15) contributed the least to the development of wheat stripe rust in the study area. This was mainly because, in 2021, the precipitation in the study area was much higher than usual for the same period. Spore invasion in wheat requires a suitable relative humidity that is sustained over a period of time; moreover, excessive precipitation can have an adverse effect, leading to the decreased importance of humidity.

*4.3. Discussion for Next Steps for Research*

This study has improved the accuracy of stripe-rust monitoring by incorporating the spatial-distribution differences of the disease. However, there are still some limitations that need to be addressed in future research. Firstly, under complex field conditions, crops are affected by multiple biotic and abiotic stressors (water, nutrients, pests and diseases, etc.), which leads to changes in physicochemical parameters [87,88]. Multi-spectral satellite data, due to their lower spectral resolution, have limited capability in distinguishing among multiple stressors. So how to utilize hyper-spectral data with higher spectral resolution to construct key vegetation traits for distinguishing multiple stressors is the focus of our next research. Secondly, this study only utilized meteorological and remote sensing data. Considering that wheat stripe rust is also influenced by multiple factors, such as topography and soil moisture, future research should incorporate more factors to improve the model's interpretability and applicability [31,82]. Additionally, this study only obtained one year of survey data for method validation. To further validate the effectiveness and applicability of this method, we plan to conduct multi-year continuous surveys in multiple wheat-rust epidemic areas in the future to obtain sufficient data to support our research. Finally, the spatial distribution information of the disease was only used during feature selection. In the future, it is worth considering incorporating this information into the monitoring model-building process. For instance, this could be achieved by adopting deep learning frameworks like Spatial_Net, where the study area could be adaptively divided into several sub-regions, and using neural network models to explain the spatial-distribution process of the disease [89].

**5. Conclusions**

Based on the spatial-distribution characteristics of wheat stripe rust in the experimental area, this study proposed a wheat-stripe-rust monitoring feature selection method that

uses the geographical detectors. The research results demonstrate that, compared to ReliefF, the top-ranking features obtained through geographical detectors exhibit a stronger discriminative ability to the disease. Simultaneously, the optimal monitoring feature set selected using geographical detectors displays a higher monitoring accuracy and stability. The findings of this study demonstrated that consideration for the spatial-distribution differences of the disease can enhance monitoring accuracy and reliability, thus providing valuable data and methodological support for the precise prevention of wheat stripe rust and the ensuring of food security.

**Author Contributions:** Conceptualization, M.Z., Y.D. and W.H.; data curation, C.R.; formal analysis, M.Z.; funding acquisition, Y.D. and W.H.; investigation, C.R.; methodology, M.Z. and Y.D.; project administration, J.G.; writing—original draft, M.Z.; writing—review and editing, M.Z., Y.D. and W.H. All authors have read and agreed to the published version of the manuscript.

**Funding:** This work was supported by National Key R&D Program of China and Shandong Province, China (2021YFB3901300), Alliance of International Science Organizations (Grant No. ANSO-CR-KP-2021-06), National Natural Science Foundation of China (32271986), SINO-EU, Dragon 5 proposal: Application of Sino-Eu Optical Data Into Agronomic Models To Predict Crop Performance And To Monitor And Forecast Crop Pests And Diseases (ID 57457), and Global Crop Pest and Disease Habitat Monitoring and Risk Forecasting(CROP PEST MONITORING), 2017–2019, 2020–2022. All authors have read and agreed to the published version of the manuscript.

**Data Availability Statement:** Data sharing is not applicable to this article.

**Conflicts of Interest:** The authors declare no conflict of interest.

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
