# Peer review of "Regional-Scale Monitoring of Wheat Stripe Rust Using Remote Sensing and Geographical Detectors"

_remotesensing, doi:10.3390/rs15184631_

Round 1

Reviewer 2 Report

Title: Remote Sensing Monitoring of Wheat Stripe Rust at aRegionalScale Based on Geographical Detectors

Comments:

The article is interesting and the described investigations are worth continuing. It will be interesting to see what the results will be from the satellite images.

1.     The purpose is actual at this time. Identification of the health of the wheat stripe is important to fix it on time. The method is clearly explained. I hope it will be of value to those interested in this topic.

2.     The introduction is well. The list of references is sufficient and new. The authors examined many sources.

3.     I did not notice any inaccuracies; the item is good quality. I recommend printing it in this form.

Author Response

Dear Reviewer,

Thank you for your positive comment on the submitted paper. We are pleased to hear that you found the content satisfactory and did not identify any specific revisions or modifications. Your encouraging comments are greatly appreciated.

If you have any further questions or if there are additional aspects you would like us to address, please do not hesitate to let us know.

Reviewer 3 Report

The paper entitled “Remote Sensing Monitoring of Wheat Stripe Rust at a Regional Scale Based on Geographical Detectors” is a very interesting work related to rapidly, non-destructively and efficaciously monitor a disease of great significance, so with important potential implications. The theme is illustrated in detail, with an almost exhaustive supporting literature, and the objectives are clear. The methodologies used are appropriate and the results are consistent. The discussion is consistent with the results and there are no speculative considerations. I encourage the publication of the article after the following revisions:

·        Introduction, second line of page 2: This what? This issue?

·        Introduction, end of the first paragraph, it would be interesting to mention the review paper “Spectroscopic detection of forest diseases: A review (1970-2020)” (J For Res 33, 21-38) in order to highlight that the proposed approach is widely used not only for Crop Pathology (mostly used), but also in Forest Pathology.

·        Introduction, the third paragraph is in accordance with the by Gongora-Canul et al. 2020. Temporal wheat blast epidemics and disease measurements using multispectral imagery (Phytopathology, 110: 393-405) focusing on another wheat disease.

·        M&M: the field survey activity was conducted only in three days, only in one year. I am wondering if this short time could represent an issue for the proposed research, as the topic is to monitor the progression of the disease, in the field. I would sugesst to provide some comments about this potential issue, at least.

·        Figure captions: Most figures should be presented with longer and more detailed captions.

·        Mentionable points for the 4.3. section, i.e., discussion for next steps for research: (1) moving from multi- to hyperspectral data (e.g., Hyperspectral assessment of plant responses to multi-stress environments: Prospects for managing protected agrosystems. Plants People Planet 2, 244-258.), and (2) predicting form (hyper)spectral data key vegetation traits related to photosynthesis and water status of crops that are commonly affected by biotic and abiotic stressors (e.g., Spectral phenotyping of physiological and anatomical leaf traits related with maize water status. Plant Physiology 184, 1363-1377).

·        Check reference style.

Minor edits required

Reviewer 4 Report

General comments:

The paper submitted for review presents the results of research focused on monitoring wheat rust disease using the Geographical detectors method.  Validation of the proposed method was performed by comparing its results with the results obtained with the help of commonly used ReliefF feature selection algorithm. The results of this comparison show that monitoring of wheat stripe rust with feature set constructed in this study was able to improve the overall accuracy and Kappa coefficient by 2.7% to 4.3% and 0.044 to 0.087, respectively in comparison to ReliefF method. The development of the proposed method is  described in detail. The whole process is also well illustrated graphically, which deserves special attention.

Generally the manuscript was prepared correctly hovewer there are some issues, which need correction before acceptance for publication.

First of all the Discussion paragraf should be expanded. The authors should refer to previous studies focusing solely on the use of vegetation indicators for strip rust monitoring and compare in detail the results presented in these studies with the results presented in the study submitted for revision. At least a few sentences about results presented by Ruan et al. in “Prediction of Wheat Stripe Rust Occurrence with Time Series Sentinel-2 Images “ published in Agriculture 2021, 11, 1079. https://doi.org/10.3390/agriculture11111079 need to be added and compare to the outcomes described in the study sent for revision.

The minore issues, which need correction are listed in Specific comments,

Specific comments:

1.     The sentence „The county, which is one of the major grain production counties in China, has a cultivated land area of 529,000 mu, with the main crop being winter wheat.” needs to be corrected. The area should be expressed in hectares.

2.     The information about the severity level mentioned in Equation (1) needs to be supplemented by more precise information. For example simething like that: The severity level of stripe rust on the leaves was divided into 9 categories (0%, 1%, 5%, 10%, 20%, 40%, 60%, 80%, and 100%).

3.     The Figure 3 right column which displays the statistical analysis of the phenological stages needs to be supplemented by adding the title of  the x axis.

4.     I am not sure whether “day of year” is the best title for the legend in Figures 3 and 4. I think that the “date” is better. Please consider this comment

General comments:

The paper submitted for review presents the results of research focused on monitoring wheat rust disease using the Geographical detectors method.  Validation of the proposed method was performed by comparing its results with the results obtained with the help of commonly used ReliefF feature selection algorithm. The results of this comparison show that monitoring of wheat stripe rust with feature set constructed in this study was able to improve the overall accuracy and Kappa coefficient by 2.7% to 4.3% and 0.044 to 0.087, respectively in comparison to ReliefF method. The development of the proposed method is  described in detail. The whole process is also well illustrated graphically, which deserves special attention.

Generally the manuscript was prepared correctly hovewer there are some issues, which need correction before acceptance for publication.

First of all the Discussion paragraf should be expanded. The authors should refer to previous studies focusing solely on the use of vegetation indicators for strip rust monitoring and compare in detail the results presented in these studies with the results presented in the study submitted for revision. At least a few sentences about results presented by Ruan et al. in “Prediction of Wheat Stripe Rust Occurrence with Time Series Sentinel-2 Images “ published in Agriculture 2021, 11, 1079. https://doi.org/10.3390/agriculture11111079 need to be added and compare to the outcomes described in the study sent for revision.

The minore issues, which need correction are listed in Specific comments,

Specific comments:

1.     The sentence „The county, which is one of the major grain production counties in China, has a cultivated land area of 529,000 mu, with the main crop being winter wheat.” needs to be corrected. The area should be expressed in hectares.

2.     The information about the severity level mentioned in Equation (1) needs to be supplemented by more precise information. For example simething like that: The severity level of stripe rust on the leaves was divided into 9 categories (0%, 1%, 5%, 10%, 20%, 40%, 60%, 80%, and 100%).

3.     The Figure 3 right column which displays the statistical analysis of the phenological stages needs to be supplemented by adding the title of  the x axis.

4.     I am not sure whether “day of year” is the best title for the legend in Figures 3 and 4. I think that the “date” is better. Please consider this comment

Round 2

Reviewer 1 Report

The authors have revised the manuscript carefully.